# Public Awareness and Sentiment toward COVID-19 Vaccination in South Korea: Findings from Big Data Analytics

**DOI:** 10.3390/ijerph19169914

**Published:** 2022-08-11

**Authors:** Yeon-Jun Choi, Julak Lee, Seung Yeop Paek

**Affiliations:** 1Department of Aviation Security Protection, Kwangju Women’s University, Gwangju 62396, Korea; 2Department of Industrial Security, Chung-Ang University, 84 Heukseok-ro, Dongjak-gu, Seoul 06911, Korea; 3Department of Criminal Justice, California State University, East Bay, SF-428, Hayward, CA 94542, USA

**Keywords:** COVID-19, vaccination, awareness, sentiment, big data analysis, South Korea

## Abstract

Despite a worldwide campaign to promote vaccination, South Korea is facing difficulties in increasing its vaccination rate due to negative perceptions of the vaccines and vaccination policies. This study investigated South Koreans’ awareness of and sentiments toward vaccination. Particularly, this study explored how public opinions have developed over time, and compared them to those of other nations. We used Pfizer, Moderna, Janssen, and AstraZeneca as keywords on Naver, Daum, Google, and Twitter to collect data on public awareness and sentiments toward the vaccines and the government’s vaccination policies. The results showed that South Koreans’ sentiments on vaccination changed from neutral to negative to positive over the past two years. In particular, public sentiments turned positive due to South Koreans’ hopeful expectations and a high vaccination rate. Overall, the attitudes and sentiments toward vaccination in South Korea were similar to those of other nations. The conspiracy theories surrounding the vaccines had a significant effect on the negative opinions in other nations, but had little impact on South Korea.

## 1. Introduction

COVID-19 has developed into a worldwide crisis since its outbreak, and vaccination has been recognized as the most effective method for controlling the spread of the pandemic along with wearing a face mask and social distancing [1]. However, many nations are facing the hurdle of citizen opposition to vaccination and “vaccine hesitancy” [2]. Experts warn that such distrust of vaccines could delay the cessation of the pandemic [3], especially in the United States where conspiracy theories regarding the vaccines have had a significant effect on people’s attitudes toward vaccination [4]. In the United Kingdom, citizens’ misunderstanding of herd immunity, fear of the unexpected, including the unknown side effects, and rumors of the vaccines being manipulated for population control have contributed to the rise of vaccine hesitancy [5]. Health risks and apprehension about the side effects such as disease reactivation, a lack of trust in policy makers, and misunderstanding of the vaccines’ effectiveness increase vaccine hesitancy which, along with conspiracy beliefs, undermines the effort to achieve herd immunity in the nations around the world [6,7].

BioNTech/Pfizer, Moderna, Oxford/AstraZeneca were among the approved vaccines [8], and most of the South Korean citizens had received a second dose when the current research was conducted. Furthermore, the Korea Disease Control and Prevention Agency (KDCA) strongly recommends a booster dose of a COVID-19 vaccine and has decreased the waiting period between doses [9]. Furthermore, the South Korean government has been considering additional booster shots [10]. However, as in the case of other nations, a small number of South Koreans have opposed the government’s vaccination policy, and disinformation regarding the vaccines has stagnated the vaccination rate [11]. For the South Korean government to improve the effectiveness of its vaccination policies, it must increase citizens’ trust in the vaccines. In order to achieve this goal, people’s awareness of and sentiments toward vaccination must be investigated. Therefore, this research explored South Koreans’ opinions on vaccination by examining online posts on the nation’s popular portals and social networking service (SNS) sites.

Specifically, this research employed big data analysis which has no set structure, making them appropriate for understanding individuals’ candid opinions and identifying their patterns [12]. Through big data analysis, we attempted to identify the most common terms and topics in online discussions during specified time periods. We also tracked the changes in the general attitudes regarding vaccination and how it varied from those of other nations. The following research questions were answered:

Research Question 1: What are South Koreans’ level of awareness and sentiment toward vaccination?

Research Question 2: How have South Koreans’ awareness of and sentiment toward vaccination changed over time?

Research Question 3: How do South Koreans’ awareness of and sentiment toward vaccination differ from those of other nations’ citizens?

## 2. Literature Review

### 2.1. Vaccination Policies and Status in South Korea

South Korea’s first confirmed cases of COVID-19 and death were reported on 19 January and 20 February 2020, respectively [13]. On 23 February, the government raised the contagious disease alert level from “warning” (orange; limited spread of disease) to “severe” (red; spread of disease in local spheres and nationwide) due to an exponential growth in the number of people contracting the disease [14]. To contain the virus, the South Korean government mandated a two-week self-quarantine for individuals who had had close contact with those who had tested positive for the disease. This measure was supplemented with testing those with symptoms to regulate suspected cases [15] by operating conventional and drive-through testing sites [16]. Through social distancing policies such as working from home, decreasing the operating hours of multipurpose facilities, reducing the maximum number of spectators allowed at festivals, and mandating face masks, the government was able to prevent the mass outbreak of COVID-19 without resorting to a lockdown. The success of the country’s vaccination policies once made it an exemplar of disease prevention and control [17] in the global community. The government eased the social distancing measures by initiating a “step-by-step recovery” policy once the country reached a 70% vaccination rate with the second dose.

South Korea started offering AstraZeneca vaccine on 26 February 2021 to individuals who were under 65 years old and were living in nursing homes as well as their staff. On 1 April 2021, vaccination for other citizens commenced. After gradually broadening the eligibility criteria, 85% of the nation’s population were vaccinated by 21 January 2022 [18,19]. After the implementation of a booster dose began on 25 October 2021, 24,505,409 South Korean residents (48% of the total population) were vaccinated over the subsequent three months [19].

### 2.2. Vaccination Sentiment and Awareness in the United States and Other Nations

Vaccination against COVID-19 in the United States began with those working in the medical field in December 2020 [20], and Pfizer, Moderna, and Janssen are the available vaccines in the country [21]. However, distrust formed among many citizens regarding the testing and approval procedures of the vaccines because of the unprecedented pace at which they were developed. In fact, conspiracy theories and apocryphal information surrounding them circulated widely on social media, which exacerbated the skepticism [22].

In a study on Americans’ vaccination hesitancy showed that the perceived possibility of contracting COVID-19 within a year, political affiliation, income level, level of education, and (not) having children at home had a moderate effect on the intention to be vaccinated [23]. The study also found that individuals supported gathering vaccine-related information from both their community and social media, regardless of whether they had a positive or negative opinion about being vaccinated. Additionally, those who do not trust their country’s medical system or health organizations have considerably weaker intentions of being vaccinated [23,24,25]. One of the main reasons for vaccine hesitancy lies in the potential side effects, which is in line with the research finding suggesting that efficacy and safety are the most important concerns when individuals are asked to choose between two fictional vaccines [25,26]. This also accords with the results of a study concerning individuals’ vaccine awareness examined prior to the distribution of the vaccines. Results found that 67% of the respondents identified themselves as “vaccine acceptant”. In the same study conducted in a country where the COVID-19 vaccines were officially authorized and began to roll out, the proportion of the vaccine-accepting individuals was 81%, suggesting an increase of positive attitudes toward vaccination [25]. As of 20 January 2022, 209 million American citizens (63% of the nation’s population) had received their second doses [18].

Regarding vaccination awareness in Canada, factors such as age, sex, and education were found to have a significant impact. Additionally, 80% of the country’s population were willing to receive vaccination once the vaccine rollout had officially begun. In the United Kingdom, a study showed that individuals who had been vaccinated against influenza were more likely to hold positive views on COVID-19 vaccination, similar to the findings in the United States [27,28]. A study from a pre-COVID-19 vaccination time in China suggests that 91.3% of Chinese people held positive views on vaccination [29].

Prior research suggests that various interrelated factors may affect people’s views and attitudes toward vaccination. However, most studies are conducted before and after the administration of first doses, so there is scant knowledge on public attitudes during the rollout of the second doses. Additionally, existing studies conducted in the cultural context of South Korea have not focused on investigating the changes in people’s opinions over different phases of the spread of COVID-19 and vaccination rollout. In order to fill this gap in the literature, we explored people’s perceptions in three phases: After the outbreak of the pandemic, before and after the first dose rollout, and the time when the second dose vaccination rate reached 70%. Text mining and sentiment analyses were employed to assess public opinions on the government’s vaccination policies and compared them to those of other nations. Drawing from the findings, we suggest strategies to improve implementation of vaccination and other relevant policies by increasing citizen confidence.

## 3. Methodology

### 3.1. Data Collection

We examined online documents and posts containing the words Pfizer, Moderna, AstraZeneca, and Janssen on Naver (www.naver.com; accessed on 3 February 2022) and Daum (www.daum.net; accessed on 3 February 2022) which are the two most popular search engines and web portals as well as Google (www.google.com; accessed on 3 February 2022) and Twitter (www.twitter.com; accessed on 3 February 2022). Pfizer, Moderna, AstraZeneca, and Janssen were chosen as the keywords since they were the available vaccines [30,31].

The research was divided into three phases, and the respective events that marked each phase were the first confirmed case of COVID-19 (19 January 2020) [32], the first date of vaccination rollout for the general public (1 April 2021) [33], and the date on which the nation’s second-dose vaccination rate reached 70% (24 October 2021) [34]. The final phase lasted until 23 January 2022.

These three events were designated as the temporal markers for the following reasons. First, the three-month period after the first COVID-19 case signified when South Korean citizens started to be directly affected by the pandemic. The three-month period before the initiation of vaccination for the general public allowed for an examination of the side effects among those who had been vaccinated earlier and various other reactions, including eagerness and concern among those soon-to-be vaccinated. The three-month period after South Korea’s vaccination rate reached 70% because was chosen because this was the point after which the COVID-19 prevention and control measures were relaxed as part of the “step-by-step recovery” program [35]. Additionally, the last phase was the time by which most South Koreans had been vaccinated, so it served as an ideal period for comparing awareness and sentiments to those in the previous phase. Each phase comprised three months because it allowed for collection of sufficient data to explore people’s concerns and reactions to the relevant events.

We utilized the web crawling feature of Textom 5.0 to collect data for each phase. This method allowed for automatic gathering of text data and analysis of morphemes [36]. Naver, Daum, and Google were selected for data collection as Nielsen Koreanclick’s website rankings listed them as the nation’s top three portals [37]. The fact that Naver and Daum were the top two most influential sites in South Korea with a combined market share of 80% [38] also contributed to their selection. Twitter was also selected because of its easily accessible data and relatively unrestricted discussion platform [39]. In sum, data for the current research were collected from blogs, webpages, news articles, and social media posts (Table 1).

### 3.2. Data Analysis

We performed data screening to eliminate data that were irrelevant to the research questions. Punctuation marks, Korean postpositions, conjunctions, and extraneous words were regarded as the stop words, while nouns, adjectives, and verbs were extracted. Additionally, words such as corona, COVID, and COVID-19 were merged into “corona” to generate meaningful analytical results. Words that had two morphemes, such as corona vaccine and corona-vaccine were treated as the same word.

Text mining followed data processing. Text mining refers to discovering information that was previously unknown by automatically extracting data from different written documents [40]. We employed the text mining method of word frequency analysis with the frequency and term frequency-inverse document frequency (TF-IDF) values serving as the evaluative indicators. Frequency values indicate the total number of times that a specific word appears in a dataset and higher TF-IDF values indicate that specific words and the documents in which they appear are more strongly related [41]. TF-IDF also gives a weighted value to high-frequency words if they are stop words, making it useful when referring to with keyword frequencies.

Moreover, topic modeling was performed after conducting a word frequency analysis. Topic modeling is an algorithm for obtaining topics from massive unstructured literature groups. It deduces topics and characteristics by clustering words with similar meanings by using context clues [42,43]. Using this method, five topics were identified and visualized using LDAvis. By offering an intertopic distance map and the top 30 most salient terms, LDAvis facilitates the understanding of the most prominent topics and the interrelationships between topics and terms [44].

We also incorporated sentiment analysis into this research to assess the overall sentiment and categorize individuals’ emotions into positive, neutral, or negative [45,46]. Specifically, we used the collected documents as training data and applied them to conduct a sentiment analysis using the Bayes classifier Words representing emotions irrelevant to the vaccines were refined or eliminated, and public attitudes toward the vaccines and the vaccination policy were examined for each phase.

## 4. Results

### 4.1. Data Analyzed for Research

As discussed, we used the names of the four vaccines administered in South Korea, Pfizer, Moderna, Janssen, and AstraZeneca, as the search words and analyzed online posts on Naver, Daum, Google, and Twitter for each phase (19 January 2020–18 April 2020, 1 January 2021–31 March 2021, and 24 October 2021–January 2022). Table 2 shows the number of documents and data volume for each source.

### 4.2. Text Mining Analysis

#### 4.2.1. Word Frequency Analysis

The word frequency analysis extracted 12,406 words from phase 1. Because phase 1 was prior to the full-scale vaccine rollout, keywords related to vaccine development such as “vaccine” (frequency: 2708; TF-IDF value: 2641.558), “development” (692; 1469.764), and “clinical trial” (436; 1144.910), and those related to the United States, including “USA” (769; 1510.062), “American stocks/shares” (447; 1388.745), and “Korean Pfizer Pharmaceutical” (1040; 2274.740) were most common.

The word frequency analysis extracted 15,101 keywords from phase 2. Because phase 2 was when the most vulnerable subsets of the population were vaccinated, “vaccine” (frequency: 15,023; TF-IDF value: 2882.651), “vaccination” (8476; 4692.116), “side effects” (1307; 2739.894) “implementation” (1114; 2389.856), “AstraZeneca” (7955; 2949.949), “Pfizer” (6230; 2666.209), “Moderna” (3613; 2992.099), and “Janssen” (2864; 2871.164) were the most prevalent words.

The word frequency analysis extracted 15,193 keywords from phase 3. Because phase 3 was when most of the South Koreans had received a second dose, and it was when third (booster) doses began to be administered, words related to the booster shot such as “booster shot” (frequency: 2111; TF-IDF value: 3652.730), “vaccination round” (2644; 4171.500), “vaccine” (9080; 5617.865), “vaccination” (9080; 5617.865), “reservation” (906; 2222.592), and “side effects” (1060; 2367.221) were most frequently searched.

#### 4.2.2. Topic Modeling

Topic modeling utilized five topics for each phase, and thirty topic words were set according to the standard keywords, Pfizer, Moderna, Janssen, and AstraZeneca (see Table 3 for the subjects drawn from the modeling). Visual analysis through LDAvis showed that the farther the distance between the topics, the more the topics were conceptually distinct. In contrast, if the distance between the topics was closer or if they overlapped, it indicated lower discriminant validity, suggesting that the topics were conceptually similar. Additionally, the size of the circle increased with the frequencies of the words it contained. Therefore, the largest circle represented the main topic.

As shown in Figure 1, the distance between the topics from phase 1 was moderately far, which indicated high discriminant validity. In phase 1, “current status of COVID-19” (topic 3) was the main topic and it was comprised of keywords such as “today”, “COVID-19 variants”, “self-quarantine”, and “confirmed cases”. Topic 1 (“the development of vaccines and cures”) was the second largest proportion and contained the words such as “cure”, “clinical”, and “research”. Topic 2 was designated as “economic issues related to vaccines” and included keywords such as “USA”, “American stock/shares”, and “dividends”. Topic 5 was named “issues related to Janssen” because of it includes words such as “vaccination round”, “Janssen”, and “additional vaccination”. Topic 4 was designated as “vaccine side effects” and “severe”, “objects of prohibition”, and “allergic reactions” were the most common words.

Topic 5 was the main topic of phase 2 and was designated as “vaccine side effects”, as it included keywords such as “side effects”, “safety”, and “symptoms” (Figure 1). Topic 5 which was the next largest proportion was named “implementation of novel vaccines” as it contained keywords such as “Novavax” and “Covax”. Topic 1 was called “issues related to vaccine procurement” with keywords including “government”, “procurement”, and “contract”. Topic 3 included keywords such as “vaccination”, “variety”, and “quarterly”, and was named “vaccination”. Topic 2’s main keywords were “virus”, “paramedics”, and “today” so was named “current status of and information on COVID-19”.

The distance between the topics from phase 3 was moderately far, indicating a high discriminant validity (Figure 1). Topic 1 was the main topic and was designated as “vaccination completion and booster shot” because it included keywords such as “vaccination”, “completion”, “booster shot”, and “additional vaccination”. Topic 5 contained words such as “Omicron” and “variant” so was designated as “variant virus issues”. Topic 3 was designated as “vaccine side effects” as it was comprised of keywords such as “worried” and “sick”. Topic 4 was named “implementation of novel vaccines” because of the words related to Chinese vaccines such as “Sinopharm” and “Sinovac”. Topic 2 was labeled as “current status of and information on COVID-19” and included words such as “today” and “changes”.

**Figure 1 ijerph-19-09914-f001:**
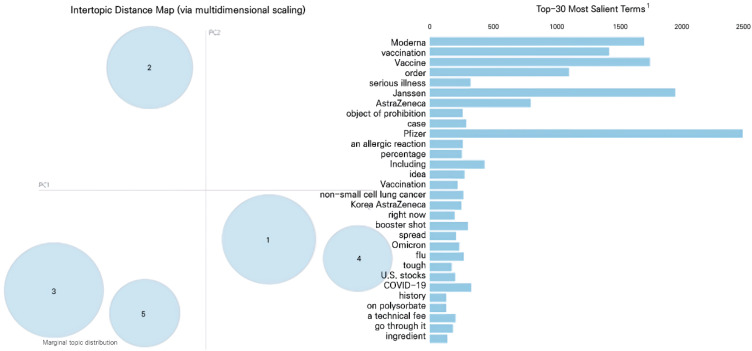
LDA topic modeling of phases 1, 2 and 3. ^1^ The figure shows thirty of the most salient terms translated from Korean into English.

### 4.3. Sentiment Analysis

The sentiment analysis for phase 1 showed that news articles reporting the onset of vaccine development consisted mostly of neutral opinions (3138/3870, 81.09%), followed by positive (14.44%) and negative (4.47%) views. As shown in Figure 2, the vaccine development process and related content evoked mostly neutral opinions. The anticipation of vaccine development and sentimental words related to it (e.g., “wanting” and “expecting”) were mostly found to be connected to positive opinions. Depression and lethargy from COVID-19 and the sentimental words related to them (e.g., “crying” and “tiring”) were mostly connected to the negative opinions.

In phase 2, concern about the side effects of vaccination contributed to a steep rise in negative opinions (1040/6706, 15.51%), with neutral and positive attitudes accounting for 74.81% and 9.68%, respectively. As shown in Figure 2, sentimental words related to the concern about the side effects (e.g., “serious” and “sick”) were mostly connected to negative opinions. Information related to vaccination was the main content corresponding to the neutral opinions, and sentimental words related to the expectation of vaccine efficacy (e.g., “good” and “keen”) were most apparent in positive opinions.

In phase 3, the proportion of positive opinions increased noticeably (1729/7113, 24.31%) compared to the previous phase, surpassing the proportion of negative opinions (21.54%). More solidified opinions and viewpoints on vaccination were also more apparent than in phases 1 and 2, as evidenced by the decrease in neutral opinions (54.15%). As shown in Figure 2, sentimental words related to the efficacy of vaccination (e.g., “good”, “okay”, and “recommended”) were mostly found in positive opinions. Content related to booster shots was associated with the neutral opinions, and the sentimental words regarding the concern about the booster shot (e.g., “concerned” and “scared”) constituted most of the negative opinions.

## 5. Discussion

The findings suggested that South Koreans’ awareness of and sentiments toward vaccination in phase 1 were mostly neutral. In addition, phase 2 featured the most negative views, and the proportion of positive opinions surpassed that of negative attitudes in phase 3. We confirmed that high expectations for vaccine efficacy and a high vaccination rate were the two contributors of increased public trust in vaccination. As prior studies have shown [11], the reason for the negative consensus could be attributed to the fear of side effects and the doubts about the vaccines’ efficacy. This may be due to the fact that the vaccines were developed and approved at an unprecedented pace as well as fearmongering about the side effects through disinformation. In fact, Twitter Korea deleted 43,000 Tweets that had shared fake news related to COVID-19, and YouTube deleted approximately 1,000,000 illegitimate COVID-19-related videos. In an effort to mitigate the impact of misinformation and disinformation on vaccinations [47].

In phase 1, the majority of the topics were related to vaccine development. This period immediately followed the first confirmed case of COVID-19 in South Korea, which explains why the main interest was the virus itself. Likewise, positive opinions on vaccination outweighed the negative ones in the sentiment analysis while neutral opinions were most common. This was when the whole world was becoming more conscious of the pandemic’s impact, resulting in growing interest in the vaccines. However, the magnitude of this awareness was expected to be marginally lower because South Korea’s confirmed cases and deaths were considerably lower compared to those of other nations [48]. This may be a result of the government’s standardized system of preventing and controlling the spread of COVID-19. Moreover, sentimental words related to depression from the pandemic were identified. One study found that 48% of South Korean citizens had experienced depression or anxiety from COVID-19 which is also called “corona blue [49]”. Based on this finding, we suggest the government implement programs such as counseling centers that can support those who experience mental health-related issues during the pandemic through counseling and other services. Moreover, promoting activities that may help enhance physical and mental health during the pandemic such as forest-bathing could be another option [50,51].

In phase 2, keywords regarding vaccination and numerous documents related to its side effects emerged. Negative opinions increased greatly while positive ones decreased compared to the previous phase during which most negative opinions were related to health. According to the Korea Internet Self-Governance Organization (KISO), most of the controversies about the vaccines’ side effects during this period were misinformation, which supports the argument that fake news about the vaccines contributed to negative opinions [52]. In the United States and the United Kingdom, conspiracy theories contributed as strongly as did the concerns about side effects to the negative views during the same period, a different pattern found compared to that of South Korea [53,54].

South Koreans were more likely to accept the messages related to public health and vaccination if they were delivered by well-known public figures [55], and a higher level of trust in the government made South Koreans less susceptible to conspiracy theories [56]. As the level of trust in the Korea Centers for Disease Control and Prevention (KCDC) which develops policies regarding the prevention and control of infectious diseases was considerably high (74% in the first week of January 2021), and the news stories and other online posts that included negative views on the vaccines usually focused on the trustworthiness of the government agencies, [57] we can infer that conspiracy theories regarding the vaccines did not have a significant effect on South Koreans’ awareness of and sentiment toward vaccination [58].

In phase 3, issues relating to booster shots, vaccination, the approval of Novavax, and the emergence of the Omicron variant constituted the majority of the identified topics. As shown in the sentiment analysis, positive opinions increased dramatically and surpassed negative ones. While some people argued that previous vaccines were ineffective at curbing the spread of the Omicron variant, there were a considerable number of positive views that a booster shot would prevent infection with the variant. As over a half the South Korean population were boosted within a three-month period and the vaccination rate was continuing to increase at a steady pace, it was expected that the rate at which people receive booster shots would continue to rise along the positive views about vaccination [19].

The awareness of and sentiment toward vaccination in South Korea shares similarities with those in other nations. The pattern of an increase in negative opinions between the start of the pandemic and the commencement of vaccination is analogous to the Korean Tweets collected between 16 March 2020, and 15 March 2021 [59]. This also holds true for the results generated from the Tweets composed in English between 28 September 2020 and 4 November 2020 [60]. Additionally, words such as “side effects”, “procurement”, and “safety” appeared most commonly, which is comparable to the most frequently discussed topics on social media in December 2020 (i.e., the time vaccination began in the United States) [53]. Most of the negative opinions in South Korea have centered on the doubt and concern about the vaccines’ safety. Similarly, most of the negative opinions regarding vaccination in the United States, Canada, and Malaysia have been shown to stem from apprehension about the vaccines’ safety [25,60,61]. Furthermore, the increase of positive opinions observed a few months after the vaccine rollout in the United States and Canada is akin to the trend found in South Korea [27].

However, differences do exist between the findings of the current research and those of the studies conducted in other nations. According to the analysis from Google Trends, the volume of the searches on prominent myths related to the vaccines and the keywords associated with them such as “coronavirus vaccine mercury” and “coronavirus vaccine autism” increased considerably in the United States [62]. An analysis of Australian Twits also found that the aluminum toxicity myth received much attention [63]. However, these keywords did not emerge in the posts composed in South Korea. Differences among the nations were especially evident because the vaccine side effects and vaccination policies were hotly debated topics in South Korea, but “vaccination for detainees” and “prioritizing vaccination for selected jobs” were the main subjects discussed in the United States [53]. In the same vein, conspiracy theories about the vaccines had a tremendous effect on the negative opinions about vaccination in the Middle East and in Western countries, while they had a negligible effect in South Korea [54,64].

In sum, the results of the current research were in accord with the findings of prior studies conducted in other cultural contexts in that negative views on the vaccines at the time of the beginning of vaccination evolved into positive attitudes as more people were vaccinated. On the other hand, one main difference was that misinformation and disinformation about the vaccines including the conspiracy theories had a negative effect on the vaccination in other countries while they had minimal impact in South Korea. As mentioned previously, it can be inferred that a high level of public trust in the KCDC lowered the perceived validity of the conspiracy theories [56,58].

## 6. Conclusions

This study identified the changes in South Koreans’ awareness of and sentiment toward COVID-19 vaccination through text mining and sentiment analysis and found that the negative consensus gradually evolved into a more positive one. It can be understood that the increased trust in the vaccines coupled with the rise of concern about infection produced positive attitudes toward vaccination [65]. However, the vaccination rate in South Korea plateaued after it surpassed the 80% mark [18], which could be attributed to a subset of the population vigorously opposing vaccination. We found the principal factors contributing to this phenomenon to be fear of the vaccines’ side effects and a lack of credible information on the vaccines’ efficacy. In particular, in accordance with existing findings, we confirmed that misinformation exacerbated individuals’ negative views on the vaccines [66,67]. The South Korean government is currently planning administration of a fourth vaccine dose, and it is expected that additional doses will follow. For such policies to be implemented effectively, the most prominent factors contributing to negative consensus—misinformation about the vaccines’ side effects and efficacy—must be eliminated. The KCDC understands the gravity of the situation so is operating a website to address the issue of misinformation related to vaccination [68]. However, the main sources of disinformation are websites and social media [5]. Therefore, positive opinions and the vaccination rate could increase if negative opinions on vaccination could be reduced by delivering genuine information through these channels [69].

Despite the potential contributions of the current research, we acknowledge the limitations resulting from not including all web portals and SNS services, thus not being able to capture certain group’s attitudes and perceptions. Therefore, the results of the current study should be interpreted with caution and may not be generalized to other settings.

## Figures and Tables

**Figure 2 ijerph-19-09914-f002:**
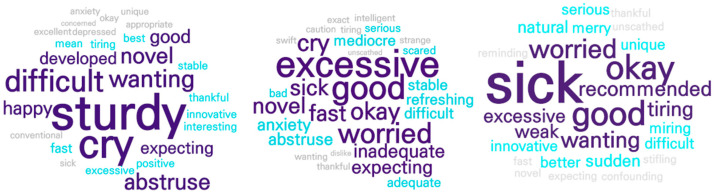
Sentiment analysis in phases 1, 2, and 3. The figure shows the sentiments.

**Table 1 ijerph-19-09914-t001:** Data Collection and Analysis Information.

Category	Content
Keywords	Pfizer, Moderna, Janssen, AstraZeneca
Periods	Phase 1 (19 January.2022~18 April 2020) Phase 2 (1 January 2021~31 March 2021) Phase 3 (24 October 2021~23 January 2022)
Channels	Naver, Daum, Google, Twitter
Software	Textom 5.0

**Table 2 ijerph-19-09914-t002:** Data Used for Research.

Data Source	Number of Documents	Volume
Naver	12,787	6749 KB
Daum	4509	1785 KB
Google	798	215 KB
Twitter	480	95 KB
Total	18,574	8884 KB

**Table 3 ijerph-19-09914-t003:** Subjects drawn from LDA topic modeling in phases 1, 2 and 3.

Phase 1			
Topic Number	Topic	Percentage	Major Keywords
3	Current status of and information on COVID-19	27.6% (1068/3870)	Today, variants, self-quarantine, confirmed cases, domestic, life
1	Development of vaccines and cures	24.5%	Cures, clinical, research, clinical trial, medicine
2	Economic issues related to vaccines	20.8%	USA, American stocks/shares, dividends, Korean Pfizer
5	Issues related to Janssen	13.9%	Rounds, Janssen, domestic, additional vaccination, standard, youth group
4	Vaccine side effects	13.3%	Severe, objects of prohibition, allergic reactions, composition, worried, death
**Phase 2**			
**Topic Number**	**Topic**	**Percentage**	**Major Keywords**
5	Vaccine side effects	27.5% (1844/6706)	Side effects, safety, symptoms, effect, clinical
4	Implementation of novel vaccines	25.8%	Novavax, Covax, coming in, launch, supply, authorized
1	Issues related to vaccine procurement	23.2%	Government, procurement, pharmaceutical company, contract, purchase, development
3	Vaccination	12.3%	Vaccination, variety, quarterly, status, hospital, people
2	Current status of and information on COVID-19	11.2%	Virus, paramedic, high-risk medical institution, today
**Phase 3**			
**Topic number**	**Topic**	**Percentage**	**Major keywords**
1	Vaccination completion and booster shot	46.2% (3286/7113)	Vaccination, completion, booster shot, additional vaccination, cross-vaccination
5	Variant virus issues	20.8%	Omicron, variant, COVID-19, infection, world
3	Vaccine side effects	14.4%	Worried, sick, problems, vaccination
4	Implementation of novel vaccines	10.1%	Sinopharm, Sinovac, KCDC, authorized, vaccines
2	Current status of and information on COVID-19	8.5%	Current, today, changes, fatality rate

## Data Availability

Not applicable.

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
