# Peer review of "Public Awareness and Sentiment toward COVID-19 Vaccination in South Korea: Findings from Big Data Analytics"

_ijerph, 2022, doi:10.3390/ijerph19169914_

Round 1

Reviewer 1 Report

This study reports on big data analysis related to the public awareness and sentiment toward COVID-19 Vaccines in South Korea.

It could be accepted after the following major revisions.

-This is a generally well-written work but there are some passages in need to be clarified. I recommend the authors to check English level of their manuscript and improve it

-COVID-19 hesitancy follows also a number of adverse effects and reactivation of different diseases widely reported in the scientific literature. Briefly mention it in the introduction and/or conclusions. See for example: https://scholar.google.com/scholar?hl=en&as_sdt=0%2C5&q=COVID+19+vaccine+disease+reactivation&btnG=

For your knowledge, see works like those with DOI: 10.3390/vaccines9091013 ;  10.1080/09273948.2021.1976221; 10.1111/jdv.17646

-Introduction: more literature examples on vaccines should be cited. Cite at least the work with DOI:10.2174/0929867328666210521164809

-line 65-66: did you mean: 'by the end of February' (without 23)?

-line 114: rewrite 'to be vaccinated was 80% after vaccination was under way.'

-line 134 onwards: provide links of all webpages visited with dates at which they were accessed

-line 153: provide more details on the software Textom 5.0 (city, country, website - if available)

-Tab 2: better explain what '± 8.6 MB' stands for. If it is a SD it looks too large. Then were are SDs for the other elements of the sum?

-line 283: did you mean 'Figure 3'? 

-probably it was better to realize figures 2 and 3 with English words (not Korean) and move the images with Korean words in a supplementary file

-line 317. Forest bathing and sport can also be suggested for protecting mental and physical health during the pandemics. Cite the works with DOI: 10.1007/s10311-021-01321-9 and 10.1007/s10311-021-01372-y

Author Response

Reviewer 1

Comments and Suggestions for Authors

This study reports on big data analysis related to the public awareness and sentiment toward COVID-19 Vaccines in South Korea. It could be accepted after the following major revisions.

- This is a generally well-written work but there are some passages in need to be clarified. I recommend the authors to check English level of their manuscript and improve it

  • Thank you for the suggestion. The authors have reviewed and revised the manuscript thoroughly to improve the quality of writing.

- COVID-19 hesitancy follows also a number of adverse effects and reactivation of different diseases widely reported in the scientific literature. Briefly mention it in the introduction and/or conclusions. See for example: https://scholar.google.com/scholar?hl=en&as_sdt=0%2C5&q=COVID+19+vaccine+disease+reactivation&btnG=

  • As suggested, we have discussed vaccine hesitancy due to adverse effects such as disease reactivation in the introductory section.

- For your knowledge, see works like those with DOI: 10.3390/vaccines9091013 ; 10.1080/09273948.2021.1976221; 10.1111/jdv.17646

-Introduction: more literature examples on vaccines should be cited. Cite at least the work with DOI:10.2174/0929867328666210521164809

  • Thank you for the suggestion. We have cited Costanzo and colleagues (2022) in Introduction.

- line 65-66: did you mean: 'by the end of February' (without 23)?

  • The South Korean government raised the COVID-19 alert level from “warning” to “severe” on February 23, 2020. We have revised the sentence to clarify the confusion.

- line 114: rewrite 'to be vaccinated was 80% after vaccination was under way.'

  • We have revised the sentence to “Additionally, 80% of the country’s population were willing to receive vaccination once the vaccine rollout had officially begun.”

- line 134 onwards: provide links of all webpages visited with dates at which they were accessed

  • We have provided the links for all webpages as well as the dates of access.

-line 153: provide more details on the software Textom 5.0 (city, country, website - if available)

  • We have provided more information on Textom 5.0 (city, country, website) in Table 1.

-Tab 2: better explain what '± 8.6 MB' stands for. If it is a SD it looks too large. Then were are SDs for the other elements of the sum?

  • We meant to show that 8,884 KB translated into about 8.6 MB. We have removed it to avoid confusion.

- line 283: did you mean 'Figure 3'?

  • That is correct. We have made the edit accordingly.

- probably it was better to realize figures 2 and 3 with English words (not Korean) and move the images with Korean words in a supplementary file

  • This is a good point. We have translated Korean into English in Figures 2 and 3. We have decided to exclude the Korean words as they were deemed unnecessary.

-line 317. Forest bathing and sport can also be suggested for protecting mental and physical health during the pandemics. Cite the works with DOI: 10.1007/s10311-021-01321-9 and 10.1007/s10311-021-01372-y

  • As suggested, we have cited Roviello and colleagues (2022) and Gilhen-Baker and researchers (2022).

Reviewer 2 Report

Thanks for the invitation to review this manuscript.

In the current study, Yeon-Jun Choi et al. investigated an important and timely topic; namely, the awareness South Koreans’ and their sentiments towards COVID-19 vaccination through different phases of the epidemic. The importance of this study is related to the resurgence of COVID-19 cases worldwide and in South Korea in summer 2022. Thus, the results of the current study can provide helpful clues to devise strategies that might help in promoting COVID-19 vaccination especially among vaccine resistant groups.

Overall, the manuscript is well-written with clear presentation of the methodology.

I have the following minor points that can help the authors to improve the manuscript:

1.      In the Introduction, the authors are recommended to expand the overview of COVID-19 vaccine hesitancy in relation to conspiracy beliefs and not to limit it to the U.S. since this phenomenon has resulted in much higher rates of vaccine hesitancy than those mentioned by the authors.

2.      In the Methodology, line 134: please be more specific regarding the websites used to gather the data.

3.      Please provide the Figure 2 in English and keep the Korean for the supplementary. The same applies for Figure 3.

4.      Please add a limitations section to address the potential caveats of the study (e.g. missing a few websites, non-representativeness of the used websites).

Author Response

Reviewer 2

Comments and Suggestions for Authors

Thanks for the invitation to review this manuscript.

In the current study, Yeon-Jun Choi et al. investigated an important and timely topic; namely, the awareness South Koreans’ and their sentiments towards COVID-19 vaccination through different phases of the epidemic. The importance of this study is related to the resurgence of COVID-19 cases worldwide and in South Korea in summer 2022. Thus, the results of the current study can provide helpful clues to devise strategies that might help in promoting COVID-19 vaccination especially among vaccine resistant groups.

Overall, the manuscript is well-written with clear presentation of the methodology.

I have the following minor points that can help the authors to improve the manuscript:

  1. In the Introduction, the authors are recommended to expand the overview of COVID-19 vaccine hesitancy in relation to conspiracy beliefs and not to limit it to the U.S. since this phenomenon has resulted in much higher rates of vaccine hesitancy than those mentioned by the authors.
  • Thank you for the suggestion. We have discussed the issue of vaccine hesitancy due to conspiracy theories and its impact on low vaccination rates worldwide in the introductory section. In addition to the example of the United States, we have discussed the cases of the United Kingdom and South Korea.
  1. In the Methodology, line 134: please be more specific regarding the websites used to gather the data.
  • This is a good point. We have provided more information on the sources of data.
  1. Please provide the Figure 2 in English and keep the Korean for the supplementary. The same applies for Figure 3.
  • We have translated Korean into English in Figures 2 and 3. We have decided to exclude the Korean words as they were deemed unnecessary.
  1. Please add a limitations section to address the potential caveats of the study (e.g. missing a few websites, non-representativeness of the used websites).
  • We have added a section on the study limitations in Conclusions.

Reviewer 3 Report

see grammatical errors and include statistical analysis

Author Response

Reviewer 3

Comments and Suggestions for Authors

See grammatical errors and include statistical analysis

  • We have revised the manuscript thoroughly and checked for spelling and grammatical errors. Also, there is no need for further statistical analyses (e.g., quantitative) since the research is based on big data analysis. Please let us know if you have any specific suggestions.

Reviewer 4 Report

The literature section is brief, but in its brevity has missed some useful recent work that is relevant to the discussion and provides context.  For example, the MDPI journal COVID has a number of recent works on vaccine hesitancy that are potentially relevant.  Also, note:

Israeli, T., Popper-Giveon, A., & Keshet, Y. (2022). Information gaps in persuasion knowledge: The discourse regarding the Covid-19 vaccination. Health. https://journals.sagepub.com/doi/full/10.1177/13634593221113208

Atkinson, C.L.; Atkinson, A.M. Vaccine Hesitancy and Administrative Burden in the Australian National Immunisation Program: An Analysis of Twitter Discourse. Knowledge 2021, 1, 25-39. https://doi.org/10.3390/knowledge1010004  (Particularly on the impact of conspiracy theory and vaccine hesitancy, given some of the material in the discussion section, and use of social media content analysis).

"Additionally, existing studies have not focused on investigating the changes in people’s opinions over different phases of the spread of COVID-19 and vaccination rollout."  Not really.  There are a variety of papers that look at trends in beliefs about COVID-19 perceptions.  Kumar, et al, 2022. Chopra, et al 2021. 

I appreciate the clear effort at segmentation of phases in 3.1

Is it possible to incorporate translation into the figures, rather than in the appendix?  In the appendix, it makes it difficult to use (back and forth to the appendix) for an international audience). See figure 2 and 3.

Around line 300 - what was the nature of this mass deletion for those that do not know about it?

The discussion/conclusion could possibly do a better job tying this analysis to existing literature and growing interpretation/theory building.

Author Response

Reviewer 4

Comments and Suggestions for Authors

The literature section is brief, but in its brevity has missed some useful recent work that is relevant to the discussion and provides context. For example, the MDPI journal COVID has a number of recent works on vaccine hesitancy that are potentially relevant. Also, note:

Israeli, T., Popper-Giveon, A., & Keshet, Y. (2022). Information gaps in persuasion knowledge: The discourse regarding the Covid-19 vaccination. Health. https://journals.sagepub.com/doi/full/10.1177/13634593221113208

Atkinson, C.L.; Atkinson, A.M. Vaccine Hesitancy and Administrative Burden in the Australian National Immunisation Program: An Analysis of Twitter Discourse. Knowledge 2021, 1, 25-39. https://doi.org/10.3390/knowledge1010004 (Particularly on the impact of conspiracy theory and vaccine hesitancy, given some of the material in the discussion section, and use of social media content analysis).

"Additionally, existing studies have not focused on investigating the changes in people’s opinions over different phases of the spread of COVID-19 and vaccination rollout." Not really. There are a variety of papers that look at trends in beliefs about COVID-19 perceptions. Kumar, et al, 2022. Chopra, et al 2021.

  • We have added the above two papers to the revised manuscript. We have also edited the problematic sentence as follows: “Additionally, existing studies conducted in the cultural context of South Korea have not focused on investigating the changes in people’s opinions over different phases of the spread of COVID-19 and vaccination rollout.”

I appreciate the clear effort at segmentation of phases in 3.1

  • Thank you.

Is it possible to incorporate translation into the figures, rather than in the appendix? In the appendix, it makes it difficult to use (back and forth to the appendix) for an international audience). See figure 2 and 3.

  • We have translated Korean into English in Figures 2 and 3. We have decided to exclude the Korean words as they were deemed unnecessary.

Around line 300 - what was the nature of this mass deletion for those that do not know about it?

  • This is a good question. The mass deletion was carried out to alleviate the negative effects of misinformation and disinformation regarding the COVID-19 vaccines and vaccination.

The discussion/conclusion could possibly do a better job tying this analysis to existing literature and growing interpretation/theory building.

  • Thank you. We have added and discussed the studies suggested by the reviewers and tied the results to the existing knowledge.

Round 2

Reviewer 1 Report

The manuscript can be published in the current form 

Author Response

Thank you!!